# Epidemiologic Analysis of Taiwanese Patients with Idiopathic Pulmonary Fibrosis

**DOI:** 10.3390/healthcare8040580

**Published:** 2020-12-21

**Authors:** Sheau-Ning Yang, Diahn-Warng Perng, Hsin-Kuo Ko, Yuh-Lih Chang, Chia-Chen Hsu, Hsin-Yi Huang, Mei-Ing Chung

**Affiliations:** 1Department of Pharmacy, Taipei Veterans General Hospital, Taipei 11217, Taiwan; snyang@vghtpe.gov.tw (S.-N.Y.); ylchang@vghtpe.gov.tw (Y.-L.C.); cchsu6@vghtpe.gov.tw (C.-C.H.); 2Department of Graduate Institute of Pharmaceutical Sciences, School of Pharmacy, Kaohsiung Medical University, Kaohsiung 80708, Taiwan; 3Department of Chest Medicine, Taipei Veterans General Hospital, Taipei 11217, Taiwan; dwperng@vghtpe.gov.tw (D.-W.P.); hkkao@vghtpe.gov.tw (H.-K.K.); 4Department of Internal Medicine, School of Medicine, National Yang-Ming University, Taipei 11221, Taiwan; 5Department & Institute of Pharmacology, National Yang-Ming University, Taipei 11221, Taiwan; 6College of Public Health, National Taiwan University, Taipei 10055, Taiwan; stathsin@gmail.com; 7Biostatistics Task Force, Taipei Veterans General Hospital, Taipei 11217, Taiwan

**Keywords:** idiopathic pulmonary fibrosis, incidence, prevalence, Taiwan

## Abstract

Several databases of epidemiologic studies in patients with idiopathic pulmonary fibrosis (IPF) have been analyzed in the Western community. However, few studies have been reported in Asia. The objective of this study was to investigate the epidemiology of IPF in Taiwan. We collected and analyzed patients with IPF from the Taiwan National Health Insurance Research Database from 2001 to 2011. We estimated the annual incidence and cumulative prevalence of IPF and mean survival time of patients and determined the causes of death. The annual incidence rates of IPF remained stable after 2005, ranging from 0.7 to 1.3 cases per 100,000 people per year, whereas the cumulative prevalence rates increased steadily from 3.1 to 6.4 cases per 100,000 people per year during 2006–2011 based on a narrow case definition. Men older than 75 years had higher incidence compared with other age groups. The mean survival after diagnosis was 6.9 years. Old age, male sex, and respiratory hospitalization were associated with shorter survival time after diagnosis. Both the incidence and prevalence rates of IPF were lower in Taiwanese patients than Western ones. Moreover, the survival time was higher in the Asian population compared with the Western population. These results may suggest the heterogeneity of the IPF definition in different study populations and geographic locations.

## 1. Introduction

Idiopathic pulmonary fibrosis (IPF) is a progressive and life-threatening interstitial lung disease (ILD). Patients usually exhibit symptoms of chronic exertional dyspnea, cough, bibasilar inspiratory crackles, and finger clubbing [1]. According to the 2011 American Thoracic Society (ATS)/European Respiratory Society (ERS) criteria [2], the diagnosis of IPF requires the exclusion of other known causes, including environmental exposure, medication, systemic disease of ILD, or the presence of a usual interstitial pneumonia pattern on high-resolution computed tomography (HRCT) of the chest in patients not subjected to surgical lung biopsy or specific combinations of HRCT and surgical lung biopsy patterns in patients subjected to surgical lung biopsy.

Although the disease is considered rare, it occurs with similar frequency to that of stomach, brain, and testicular cancers [3]. The incidence of IPF has risen over time and is estimated to range between 2.8 and 18 cases per 100,000 persons per year in Europe and North America [3,4]. Increasing rates of hospital admissions and deaths due to IPF also suggest an increasing burden of the disease [4,5,6].

Most epidemiologic studies [7,8,9,10,11,12] of IPF focused on Western countries, including the United States, United Kingdom, Denmark, and Italy. However, few studies have investigated this disease in Asia. In particular, the outcome after hospitalization has not been well characterized in Taiwan. Hutchinson et al. [3] found that a few studies from Asia used insurance and claims databases, and all reported lower incidence rates than other regions, ranging from 1.2 to 4.16 per 100,000 persons per year [13,14]. Lai et al. [13] reported that the severity of IPF and survival are higher and lower in Taiwan than in other countries, respectively, suggesting that milder cases were not being detected. This study used a large database containing a large study sample, but it was not able to collect detailed clinical data. As the pharmacologic therapy for IPF improves, understanding the epidemiology of IPF will allow policy-makers to develop policies to facilitate the care of these patients. The objective of the present study was to investigate the epidemiology of IPF in Taiwan between 2001 and 2011 by using the National Health Insurance Research Database (NHIRD).

## 2. Materials and Methods

### 2.1. Study Design

This is a retrospective cohort study based on analysis of the database from the NHIRD. The National Health Research Institutes (NHRI) organizes the NHIRD, which contains all claims data from the Bureau of National Health Insurance (BNHI). To protect patient privacy, all patients were scrambled by BNHI. The NHI program covers almost the entire population of Taiwan. Therefore, the NHIRD can be used as a suitable research resource for analyzing the epidemiology of IPF in Taiwan. We identified patient deaths from the registration and hospital admission files. The code of death or automatic discharge is shown in the hospitalization data, and the record of the patient’s withdrawal from the health insurance is shown in the registration file. We consulted with the institutional review board of Taipei Veterans General Hospital (approval number: IRB 2016-06-004CC), which waived the informed consent for our retrospective observational cohort study.

### 2.2. Data Source

The Taiwan NHRI manages the NHIRD for research purposes. The National Health Insurance (NHI) program of Taiwan provides comprehensive medical services for all Taiwanese citizens, and it has been operational since 1995 [15]. The NHIRD contains the claims data of 23 million individuals, which provides coverage to >99% of the entire population of Taiwan. The NHI medical claims databases contain information on demographic characteristics, disease diagnosis, treatment and related medical expenditures, and orders of ambulatory and inpatient care. This study used claims data from the Longitudinal Health Insurance Database 2000 (LHID2000) and LHID2005, which contain all the original claims data of 1 million beneficiaries randomly sampled from the NHIRD from 1995 to 2011. The released database has been validated by the NHRI to be representative of the total population of Taiwan. The insurance claims data of Taiwan were validated for accuracy and had 92% accuracy for patients with ≥1 hospitalization in a year [16]. The data between 1 January 2000 and 31 December 2011 (study period) were analyzed, and health plan enrollees who were 0–120 years old during the study period were included in the analyses.

### 2.3. Study Sample

In this retrospective cohort study, we analyzed the epidemiology of IPF in Taiwan. We enrolled adult patients (≥18 years old) who were diagnosed with IPF from 2001 to 2011. The inclusion criteria were patients diagnosed with IPF with at least one recorded outpatient visit, emergency room visit, or hospitalization for IPF. Patients with IPF were defined according to the International Classification of Diseases, Ninth Revision, Clinical Modification (ICD-9-CM) code 516.3.

We consulted with Raghu et al. [9] to clarify the broad and narrow case definitions of IPF. The broad case definition consists of the following conditions: (1) the patient’s age was 18 at the time of confirming IPF incidence; (2) there was at least one NHI claim with IPF diagnosis (ICD-9-CM code: 516.3) between 1 January 2001 and 31 December 2011; and (3) for at least one NHI claim with IPF diagnosis, there was no NHI claim with the diagnosis for any other ILDs (Appendix A, ICD-9-CM Codes for Interstitial Lung Disease Other Than Idiopathic Pulmonary Fibrosis) since the day of IPF diagnosis (the time of confirming IPF incidence was the date of the first NHI claim meeting this condition). The narrow case definition consists of the following conditions: (1) all conditions for the broad case definition were satisfied; (2) patients with at least one NHI claim with IPF diagnosis should have at least one NHI procedure for surgical lung biopsy, transbronchial lung biopsy, or computed tomography of the thorax (Appendix A, NHI codes for Procedure of Diagnosing Lung Disease) before or on the day of IPF diagnosis. Hospitalizations of at least 1 day in duration subsequent to IPF diagnosis were categorized by ICD-9-CM as respiratory or non-respiratory in nature. The ICD-9-CM for respiratory causes is shown in Appendix A. The Taiwan NHI started to use ICD-9-CM completely in 2000 and switched to ICD-10-CM in 2016. Therefore, the patients were always diagnosed using ICD-9-CM in 2001–2011. The participants were followed until death, withdrawal from the NHI program, or the end of the study (31 December 2011), whichever occurred first.

### 2.4. Statistical Analysis

We counted the number of patients diagnosed with IPF in each year during 2001–2011 and calculated the incidence rate of a year as the number of newly diagnosed patients per 100,000 people ever alive in the NHI system in the year. The prevalence rate was measured as the number of patients per 100,000 people alive in the NHI system as of the end of a year. We performed the analyses separately for different age–gender groups on the basis of the broad and narrow case definitions. We measured the survival time (in years) for patients with incident IPF from the start of the index day to the date of all-cause death or the end of 2011, whichever came first. The results are shown as mean ± SDs for continuous variables and percentage for categorical and ordinal variables. We estimated the time (with 95% confidence interval [CI]) on the basis of a Kaplan–Meier survival estimator [17], with results stratified by subgroups of age, sex, and hospitalization. The differences between the curves were assessed by the log-rank test. For all tests, *p* < 0.05 was considered significant. All analyses were performed using SPSS version 19.0 (IBM, Armonk, NY, USA) and STATA 12 (StataCorp LP, College Station, TX, USA) statistical software.

## 3. Results

### 3.1. Study Population

The total population in Taiwan is approximately 23 million. Given that the database sampling is about 1:11, approximately 2 million men and women aged 0 years and older participated in the study database that came from LHID2000 and LHID2005. A total of 1,916,514 individuals were eligible from this database. Duplicate patients were deleted from the registration dataset (ID), and 348 IPF patients aged 18 years with ICD-9 code 516.3 were obtained and evaluated between 2001 and 2011. After the exclusion of 49 patients with other codes for ILD, there were 299 patients with IPF incident by broad case definition, and after the exclusion 177 patients without the ICD code with NHI procedure for surgical lung biopsy, transbronchial lung biopsy, or computed tomography of the thorax (Appendix A) before or on the day of IPF diagnosis, there were 122 patients with IPF incident by narrow case definition. Among these patients, 2 were excluded as the follow-up lasted less than 1 month, 25 patients were classified as respiratory hospitalizations, and 95 patients were classified as non-respiratory hospitalizations or had never been hospitalized. Finally, a survival analysis was conducted according to the patient’s age, gender, and hospitalization (Figure 1). The mean age was 65.0 ± 16.7 years, and the male-to-female ratio was 2.4:1.

### 3.2. Annual Incidence and Cumulative Prevalence Rates

Based on the narrow case definition, the annual incidence of IPF ranged from 0.2 to 1.3 new cases per 100,000 person-years in 2001–2011, and the cumulative prevalence of IPF increased from 0.2 per 100,000 persons in 2001 to 6.4 per 100,000 persons in 2011 (Table 1). Based on the broad case definition, the annual incidence of IPF ranged from 0.8 to 1.9 new cases per 100,000 person-years in 2001–2011, and the cumulative prevalence of IPF increased from 0.8 per 100,000 persons in 2001 to 15.6 per 100,000 persons in 2011 (Table 1). Both prevalence and incidence increased yearly, whether by year or by broad or narrow case definition.

### 3.3. Comparison of Incidence and Prevalence Rates among Age–Gender Groups

Based on the 2011 data, the prevalence and incidence of IPF by age and gender are shown in Table 2. For broad case definition, the age-specific incidence of IPF was higher for males than for females in groups older than 65 years. The highest incidence for males and females was in the 75–84 (12.5 per 100,000 population) and over 85 age groups (6.2 per 100,000 population), respectively. The age-specific cumulative prevalence was also higher in males than in females in all age groups. The age group with the highest prevalence was the same (75–84 age group) for males (132.5 per 100,000 population) and females (55.3 per 100,000 population). For the narrow case definition, the age-specific incidence of IPF was higher for males than for females in groups older than 65 years old. The highest incidence for males and females was in the 75–84 (7.2 per 100,000 population) and over 85 age groups (3.1 per 100,000 population), respectively. The age-specific cumulative prevalence was also higher in males than in females in all age groups. The age group with the highest prevalence was the same (75–84 age group) for males (51.9 per 100,000 population) and females (19.8 per 100,000 population). Both prevalence and incidence were generally higher among the elderly and males.

### 3.4. Survival Analysis and Causes of Death

The mean year from IPF diagnosis to death or the last follow-up was 6.9 years (95% CI: 6–7.1 years). The results of survival analysis by gender, age, or hospitalization are presented in Figure 2 and Figure 3. The survival analysis showed that the survival time was lower in males than in females (6.7 years (95% CI: 5.8–7.6) vs. 8.5 years (95% CI. 6.9–9.9); *p* = 0.122) (Figure 2A) and in individuals older than 65 years than those younger than 65 years (6.9 years (95% CI: 5.7–8.0) vs. 8.1 years (95% CI: 6.9–9.2); *p* = 0.089) (Figure 2B). However, no statistically significant difference was observed between gender and age (Figure 2). The mean survival year was lower in patients with respiratory hospitalization than in those non-respiratory hospitalization (3.6 years vs. 7.6 years; *p <* 0.001) (Figure 3). According to the narrow case definition, 32 (22.6%) patients died during the 11-year observation period, 4 patients (3.3%) died during their hospitalization, and 13 (10.6%) patients died within 1 year after hospitalization. Most of the primary cause of death was the progression of IPF in different conditions (Table 3).

## 4. Discussion

During the study period, the diagnostic criteria to identify persons with IPF were lacking. We consulted with Raghu et al. to clarify the narrow and broad case definitions of IPF. Because our narrow case finding criteria and the case ascertainment methods used by Raghu et al. [9] probably reduced the sensitivity to improve specificity, the results based on these approaches may have underestimated the true extent of the disease. For this reason, we also report estimates based on more liberal criteria as a broad case definition.

In our study, we used the large nationally representative Taiwan database to investigate the epidemiology of IPF in Taiwan. According to a previous study [18], the healthcare database is one of the most common methodologies used to collect cases of IPF. In the present study, the cumulative prevalence of IPF varied from 0.8 to 15.6 and 0.2 to 6.4 per 100,000 persons based on the broad and narrow case definitions, respectively. The annual incidence varied from 0.8 to 1.9 and 0.2 to 1.3 per 100,000 persons according to the broad and narrow case definitions, respectively. The reported values refer to the study period between year 2001 and 2011. Both the prevalence and annual incidence increased steadily from 2001 to 2011. IPF is difficult to diagnose with certainty. However, diagnostic criteria have evolved in recent years. Moreover, the use of HRCT as the diagnostic tool for IPF has become more available in Taiwan than it was 10 years ago. In fact, based on the 2011 ATS guidelines, the diagnosis of IPF requires the exclusion of known causes of ILD and the existence of the usual interstitial pneumonitis pattern on an HRCT scan in patients whose surgical lung biopsy is not indicated. Therefore, the prevalence and incidence rates increased during the study period.

Comparing our study findings with other studies is hard due to the heterogeneity of IPF definition and in different study population, the time period of analysis, and geographic locations. In North America, Raghu et al. evaluated the incidence and prevalence of IPF by analyzing the medical claims database [9]. Although they used the same case definition, the estimated incidence rate was 6.8 and 16.3 per 100,000 person-years during 1996–2000 by narrow and broad case definitions, respectively; however, the estimated incidence increased to 15.9–31.1 and 31.3–43 per 100,000 person-years in 2001–2011 [19]. The prevalence of IPF was 14.7 and 42.7 based on their narrow and broad case definitions during 1996–2000, whereas it was 15.9–31.1 and 82.6–233.3 per 100,000 person-years based on their narrow and broad case definitions in 2001–2011, respectively. Only patients aged >65 years were assessed in the later study.

In Europe, three different research groups studied the incidence and prevalence of IPF by analyzing the nationwide primary care database. Nonetheless, the broad and narrow case definitions that were used in those studies included slightly different criteria. Navaratnam et al. [8] reported that the crude overall incidence rate was 7.44 per 100,000 person-years during 2000–2008 based on the code of IFA/CFA/PF. In the study of Kornum et al. [12], which focused on the epidemiology of IPF during 2001–2005, the crude incidence rate was 5.28 per 100,000 person-years based on ICD-10 J 84.1. Harari et al. [11] investigated the epidemiology of IPF in Italy, which was based on code ICD-9-CM 516.3 during 2005–2010, and the results showed that the estimated incidence rate was 2.3 per 100,000 person-years.

In East Asia, four studies used the insurance and claims databases, and all studies reported crude incidence rates between 1.2 and 4.6 per 100,000 persons per year. Our results suggested that the incidence might be similar to those estimated in other Asian countries. Furthermore, Lai et al. [13] reported that the incidence rate was 0.2–1.3 per 100,000 persons per year from 1997 to 2007 in Taiwan. Although the incidence and prevalence rates increased throughout the study period, they were still lower than those in European countries and the United States. The lower incidence in Taiwan compared with that in other regions may be because the present study used Raghu et al. [9]’s algorithms to formulate definitions for IPF, which were stricter than others or underdiagnosed. However, we could not exclude that the healthcare system does not have sufficient support to identify cases during the study period, ethnic differences, or a combination of these factors.

The disease awareness of patients and physicians was expected to improve after the healthcare system released the following policies: Evidence-based guidelines for the diagnosis and management of IPF established by the Taiwan Thoracic Medical Association in 2015, which allows readers to understand the definition of IPF and the latest diagnostic criteria and treatment strategies, enhance physicians’ diagnostic vigilance and ability and show appropriate treatments to enhance patient well-being. In 2017, the BNHI launched a new classification system for medical treatment. The system encouraged patients with minor symptoms to seek treatment from small clinics and those with serious illnesses to go to major hospitals. In this system, the primary care physician advises patients with known or suspected IPF to seek a center with expertise in IPF care because delayed access is independently associated with increased risk of death. The referral provides patients with access to expertise in diagnosis and management, including initiation of disease-modifying therapy, monitoring, side-effect control, and non-pharmacological support.

In our study, the incidence of IPF in males was 2.4-fold higher than that in females. This rate was higher than those reported by Raghu et al. [19] in the United States and Harari et al. in Italy [11]. Both studies reported that the incidence in males is 1.3-fold higher than that in females. According to our study, the average age of onset was 65 years, which was younger than that of other Western countries [1,20]. Although the prevalence and incidence rates were not adjusted by age and gender, the incidence rate was the highest in 2011 when analyzing the annual prevalence and incidence rates during the study period. In this year, a stratified analysis of age and gender was performed; however, both prevalence and incidence increased with age, and the highest incidence was at age >75 years despite sex or case definitions, which is similar to previous studies [11,21].

The mean survival time was 6.9 years from diagnosis to the end of the study. This survival time varied with sex and age at diagnosis; females and patients <65 years of age seemed to have longer survival time (8.5 and 8.1 years, respectively). The median survival time was longer than those of previous studies in Western (3–5 years) [2] and Asian populations (0.9–2.9 years) [13,21]. However, our findings were supported by results from previous studies [5,22] that have linked younger age to increased survival time. Furthermore, our analysis showed that patients aged 60–69 years have an 8-year survival time, which is consistent with Raghu et al. [19]. Previous estimates of survival time in IPF have been derived from studies with smaller cohorts and patients receiving tertiary care, which could explain why historically estimated survival times were poor [14,23]. The outcome of respiratory hospitalization was poorer compared with that of non-respiratory hospitalization (3.6 years vs. 7.6 years; *p* < 0.001). Our finding is similar to that of Brown et al. [22], who reported that all-cause hospitalizations were strongly associated with mortality at a tertiary center.

We hypothesized that a better survival outcome will be anticipated if the case data were homogeneously collected. Lai et al. [13] reported that the median survival time was 0.9 years in Taiwan during 1997–2007 using the NHIR database, whereas our investigation results showed that the mean survival time was 6.9 years. Lai et al. investigated the epidemiology of IPF by using a database from patients who were treated with invasive or noninvasive respiratory care in NHI, suggesting that milder cases were not being reported. This study was analyzed the Longitudinal Health Insurance Database; however, no significant difference in age, sex, number of births per year, and the average insured amount was observed compared with the original NHIRD. Moreover, Lai et al. found that the all-cause mortality rate in patients with narrow case definition throughout the study period was 52% (286/550), whereas the mortality rate in our study was 26% (32/122). This observation could explain why the severity of IPF and survival in Lai et al.’s study were higher and lower than those in our study, respectively.

In the past, the 2011 ATS/ERS practice guideline [2] did not recommend the use of any specific treatment for patients with IPF. However, after the pathogenesis of IPF was known, the treatment has shifted from anti-inflammatory drugs to disease-modifying drugs. In recent years, two medications have been shown to be safe and effective in the treatment of IPF, and both are recommended for use in patients with IPF. Taiwan approved the marketing of these drugs in 2017. Because our study period is from 2001 to 2011, the impact of the use of these drugs on patients could not be determined. We look forward to an epidemiological analysis of related drugs in the future.

Lung transplantation can prolong the lifespan of IPF patients. Taiwan has had successful lung transplants since 1991. IPF is also a waiting list for indications, but there are few organ donors and not many cases of lung transplantation. Therefore, IPF lung transplants are even rarer compared with other cases.

The primary causes of death in IPF are the progression of IPF (41%), followed by cerebral arterial (13%) and cardiovascular diseases (9%) (Table 1). Our findings are consistent with those of a previous study, wherein the progression of IPF was the primary cause of death (47%), followed by cardiovascular (20%) and cerebral vascular diseases (6%) [20]. In Asia, the results were different. In Japan, Natsuizaka et al. [21] reported that the primary causes of death in IPF were IPF progression (64%), followed by lung cancer (11%), pneumonia (7%), and cardiovascular diseases (3%). In Taiwan, our results are similar with those of Lai et al. [13], who reported that IPF progression caused 46% of deaths, followed by cancer (11.2%) and cardiovascular diseases (10.1%). This discrepancy may be due to ethnic differences.

Our study has the following strengths. First, we examined the epidemiology of IPF in Taiwan based on NHIRD samples, which include a large representative population, and the accuracy of insurance claims data in Taiwan was 92% for patients with ≥1 hospitalization per year [16]. Second, the identification of IPF cases as broad or narrow cases was based on the algorithms of Raghu et al. [9], which were stricter than those reported in previous studies. Third, the NHIRD contained insurance claims data from outpatients and inpatients. This approach had the advantage of allowing the identification of patients who have never been admitted to the hospital due to IPF. Furthermore, milder cases were identified in this study.

Despite the findings, this study has some limitations. First, patients with IPF were identified based on their access to healthcare service, and determining the exact time of IPF onset was impossible. Therefore, the date of the first encounter with inpatient or outpatient healthcare systems coincided with the first recorded medical diagnosis. Second, the case definition used to identify patients with IPF in this study was not validated with respect to medical charts, and whether or not the diagnoses were based on multidisciplinary discussions was unknown. Third, NHIRD lacked detailed clinical and laboratory data; therefore, we could not identify the severity of patients at first diagnosis, which might influence the explanation of the study results.

## 5. Conclusions

The incidence and prevalence rates of IPF in Taiwanese patients were lower than those in Western communities. Men aged 75 years and older comprised the major group suffering from IPF. The survival time in the present study was higher than those of previous studies in the Asian population. These results may suggest the variety of IPF definitions and differences in the study population, time period of analysis, and geographic locations. In light of the rising prevalence and incidence of IPF, we hope that the next 5 years will be marked by greater recognition of the disease on the part of primary care providers, leading to earlier involvement of a multidisciplinary team to aid in diagnosis and management.

## Figures and Tables

**Figure 1 healthcare-08-00580-f001:**
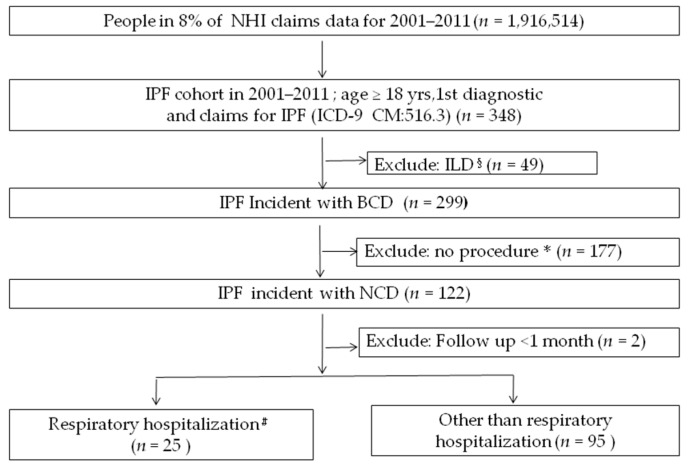
A flow chart to identify the cohort from Taiwanese NHIRD. ^§^, ICD-9-CM for ILD; *, ICD-9-CM for lung biopsy or computed tomography of the thorax; ^#^, ICD-9-CM for respiratory hospitalization. BCD, broad case definition; ILD, interstitial lung disease; IPF, idiopathic pulmonary fibrosis; NCD, narrow case definition; NHIRD, National Health Insurance Research Database.

**Figure 2 healthcare-08-00580-f002:**
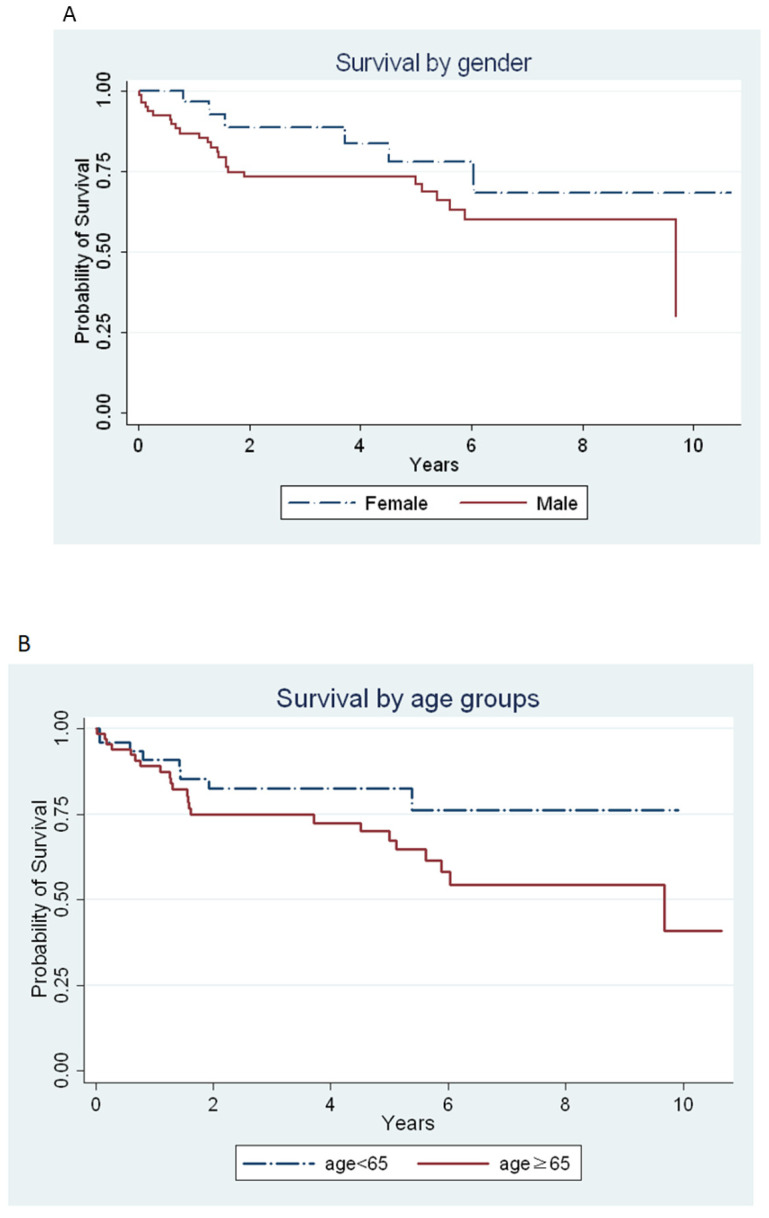
The survival of the entire cohort from initial diagnosis to death, last follow-up, or at the end of the study period. (**A**) A comparison between females and males (mean survival, 8.5 years vs. 6.7 years; *p* = 0.122). (**B**) A comparison between individuals younger than 65 years and those older than 65 years (mean survival, 8.1 years vs. 6.9 years; *p* = 0.09).

**Figure 3 healthcare-08-00580-f003:**
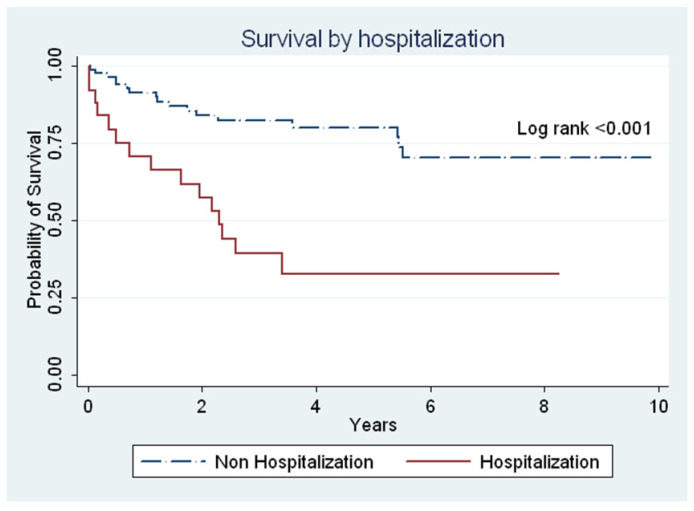
The survival of the entire cohort from initial diagnosis to death, last follow-up, or at the end of the study period. A comparison between patients with respiratory hospitalization and those with non-respiratory hospitalization (mean survival, 3.6 years vs. 7.6 years; *p* < 0.001) is given.

**Table 1 healthcare-08-00580-t001:** Prevalent and incident cases of IPF from 2001 to 2011 by case definition.

Prevalent Cases					Incident Cases			
	Narrow Case Definition		Broad Case Definition		Narrow Case Definition		Broad Case Definition	
Year	*n*	Prevalence Rate ^1^	*n*	Prevalence Rate ^1^	*n*	Incidence Rate ^2^	*n*	Incidence Rate ^2^
2001	4	0.2	15	0.8	4	0.2	16	0.8
2002	11	0.6	38	2	8	0.4	22	1.2
2003	24	1.3	57	3	12	0.6	19	1
2004	41	2.2	84	4.5	18	1.0	28	1.5
2005	46	2.4	108	5.6	4	0.2	24	1.2
2006	60	3.1	141	7.3	14	0.7	31	1.6
2007	69	3.6	169	8.8	10	0.5	30	1.6
2008	83	4.3	200	10.4	13	0.7	31	1.6
2009	92	4.8	234	12.2	10	0.5	34	1.8
2010	98	5.1	263	13.7	4	0.2	27	1.4
2011	122	6.4	299	15.6	25	1.3	37	1.9

^1^ The prevalence rate was measured as the number of patients per 100,000 persons alive in the NHI system at the end of a year. ^2^ The incidence rate was measured as the number of newly diagnosed patients per 100,000 persons ever alive in the NHI system in a year.

**Table 2 healthcare-08-00580-t002:** Prevalent and incident cases of IPF in 2011 by age, sex, and case definition.

Prevalent Cases					Incident Cases			
	Narrow Case Definition		Broad Case Definition		Narrow Case Definition		Broad Case Definition	
Year	*n*	Prevalence Rate ^1^	*n*	Prevalence Rate ^1^	*n*	Incidence Rate ^2^	*n*	Incidence Rate ^2^
Men								
Age, year								
18–34	6	2.6	8	3.4	1	0.4	1	0.4
35–44	6	3.6	8	4.8	2	1.2	3	1.8
45–54	6	3.6	13	7.9	1	0.6	1	0.6
55–64	15	12.3	28	22.9	3	2.5	4	3.3
65–74	19	29	47	71.8	4	6.1	5	7.6
75–84	29	51.9	74	132.5	4	7.2	7	12.5
85+	5	14	20	56.1	2	5.6	4	11.2
Women Age, year								
18–34	3	1.2	6	2.4	0	0	0	0
35–44	4	2.4	7	4.2	2	1.2	2	1.2
45–54	1	0.6	9	5.6	0	0	1	0.6
55–64	9	7.4	22	18.1	3	2.5	6	4.9
65–74	7	10.3	24	35.2	2	2.9	2	2.9
75–84	10	19.8	28	55.3	0	0	2	4
85+	2	6.2	5	15.4	1	3.1	2	6.2

^1^ The prevalence rate was measured as the number of patients per 100,000 persons alive in the NHI system at the end of a year. ^2^ The incidence rate was measured as the number of newly diagnosed patients per 100,000 persons ever alive in the NHI system in a year.

**Table 3 healthcare-08-00580-t003:** Primary causes of death for different conditions.

Causes of Death, *n* = 32	In Hospital	<1 Year after Hospital	Total
Progression of IPF Acute respiratory failure	3 (75)	5 (38.4)	12 (37.5)
Chronic respiratory failure		1 (7.7)	1 (3.1)
Pneumonia			2 (6.3)
Respiratory system cancer		1 (7.7)	1 (3.1)
Pneumoconiosis			1 (3.1)
Sepsis			1 (3.1)
Spontaneous pneumothorax		1 (7.7)	1 (3.1)
Cerebral arterial disease Subarachnoid hemorrhage	1 (25)	1 (7.7)	2 (6.3)
Intracranial injury of other and unspecified nature		1 (7.7)	1 (3.1)
Cerebral artery occlusion		1 (7.7)	1 (3.1)
Coronary atherosclerosis		1 (7.7)	3 (9.4)
Small intestine, including duodenum		1 (7.7)	1 (3.1)
DM with polyneuropathy			1 (3.1)
Urinary tract infection			2 (6.3)
Others			2 (6.3)
Total	4 (100)	13 (100)	32 (100)

IPF, idiopathic pulmonary fibrosis; DM, diabetes mellitus.

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
