# Peer review of "Epidemiologic Analysis of Taiwanese Patients with Idiopathic Pulmonary Fibrosis"

_healthcare, 2020, doi:10.3390/healthcare8040580_

Round 1

Reviewer 1 Report

In this manuscript by Yang et al., the authors present an epidemiological analysis of Taiwanese patients with idiopathic pulmonary fibrosis (IPF).

In summary, the data presented within the manuscript does not provide any significant new insight into IPF in Taiwanese patients. In Table 1, the sample number of patients utilized is too small to have meaningful significance.

The authors have highlighted a number of limitations of this study in the discussion section of their manuscript including: (1) it was not possible to determine the exact time of onset of IPF in individuals due to their access to Healthcare, (2) the definition used to identify IPF patients was not validated and (3) access to detailed clinical and laboratory data for IPF patients was limited and thus, therefore it was not possible to determine the severity of IPF patients ay their diagnosis. These limitations severely impact upon the significance of the data presented in this manuscript.

There are typographical errors throughout the manuscript. Erroneous use of spacing between words and punctuation occurs throughout. The English language and style content are fine.

Author Response

We would like to thank the editor and reviewers for their extensive review of our manuscript, as well as their important and helpful comments and suggestions. We provide our answers to certain comments below. The revised portions of the manuscript are highlighted in red color in our manuscript. We hope that the changes made will be considered satisfactory.

Point 1: In summary, the data presented within the manuscript does not provide any significant new insight into IPF in Taiwanese patients. In Table 1, the sample number of patients utilized is too small to have meaningful significance.

Response 1: We apologize for the misunderstanding. Table 1 indicates the primary causes of death in IPF patients. The broad and narrow case definition subgroups included 299 and 122 patients with incident disease, respectively, from January 2001 to December 2011. The patient samples are shown in the flow chart to identify cohorts from the Taiwanese NHIRD (Figure 1).(page 4, lines 141-144)

Point 2: The authors have highlighted a number of limitations of this study in the discussion section of their manuscript including: (1) it was not possible to determine the exact time of onset of IPF in individuals due to their access to Healthcare, (2) the definition used to identify IPF patients was not validated and (3) access to detailed clinical and laboratory data for IPF patients was limited and thus, therefore it was not possible to determine the severity of IPF patients ay their diagnosis. These limitations severely impact upon the significance of the data presented in this manuscript

Response 2: Thank you for your careful review and valuable comments.

We agree that it is difficult to determine the exact time of onset of IPF in individuals and the severity of IPF patients. In this study, we could only discuss the impact of age, gender, and hospitalization on survival. In our submitted manuscript for another journal, the prognostic factor for hospitalized patients with IPF included higher GAP score, lower oxygenation, higher C-reactive protein levels, higher neutrophil counts, and lower serum albumin levels.

Point 3:There are typographical errors throughout the manuscript.

Response 3: All typographical errors have been examined and corrected.

Point 4: Erroneous use of spacing between words and punctuation occurs throughout.

Response 4: We think that the erroneous use of spacing between words and punctuation may be attributed to the transformation of the document. It is now corrected.

Point 5: The English language and style content are fine.

Response 5:

We have sent the manuscript to an academic editing service again to improve the quality and fulfill the standard for publication.

Reviewer 2 Report

The article aims at describing the epidemiology of IPF in Taiwan, using data from a large national administrative database. The research question is of interest as the epidemiology of IPF may vary across countries.

The major criticism to this article is the poor quality of written English, which needs extensive review. Therefore, the understanding of many parts of the article is very difficult.

In addition, the manuscript could greatly benefit from the clarification of the following important points that need to be better addressed by Authors.

Abstract:

Line 21: Causes of deaths are not “estimated”

Line 24: the meaning of “narrow definition” is reported in the article, but never mentioned in the abstract; therefore, the reader cannot understand this definition. I suggest indeed using  “narrow case definition”

Bibliographic references in the text are not correctly displayed. Put them before the end of the sentence.

Line 53-54: please rephrase and move the sentence to either the Materials and methods section or to the Discussion section, depending on its reformulation

  1. Materials and methods

It is not clear how the LHI database is generated from the NHI database and for which purposes (research ones?). Please describe the NHI database in terms of data sources, included data and patients’ identification. The sampling process through which the LHID is generated should be better described.

It is not clear how mortality data are collected. Do they derive from the death national registry or mortality data are referred to deaths occurring during hospitalization?
In the first case, please describe the source of data used and the record-linkage process used to trace patients’ pathways.

Considering the study period, patients’ diagnoses were always coded using ICD9-CM or in some years ICD-9 has been used?

Line 77-78: Please justify with a bibliographic reference the data provided  on insurance data validation.

Study sample

Please describe the method used for classifying the hospitalization in respiratory or non- respiratory.

Please provide bibliographic reference for the Kaplan-Meyer survival method.  

Results

Figure 1. Please reformulate it according to the comments about the sampling procedure

n=488 is not IPF prevalence, n=299 does not represent IPF incidence. They are rather the numerators used to calculate prevalence and incidence.
# is not explained in the legenda

I think this section could benefit form the adding of a table  showing  prevalence and incidence rates of IPF from 1997 to 2007, by case definition (narrow vs broad)

Prevalent cases

Incident cases

years

Narrow case def

Broad case def

Narrow case def

Broad case def

2001

N

Prevalence rate

N

Prevalence rate

N

Incidence rate

N

Incidence rate

   Discussion

Line 262: please add that data are referred to the study period and that reported values refer to year 2001 and 2011 respectively.

General comments

The discussion would greatly benefit from a general reorganization.

It should be focused around the main findings of this study that are:

  • The increasing prevalence and incidence rates during the study period, using both a narrow and the broad cease definition
  • Prevalence and incidence rates in Taiwan are lower than in Western countries. Does it depend on the definitions used, the ability of the system to identify cases, ethnic differences or a combination of these factors?
  • Regarding the ability of the system to identify cases, a brief description of health policies in Taiwan and RD policies could help
  • Age and sex as modifier effects
  • Survival: in this study was significantly higher than ina rpevous study carried out in Taiwan [Lai et al.]. Authors should try to explain this difference.
  • Patents’ access to new drugs for IPF entered in the market during the study period should be discussed, specifying if these treatments are accessible in Taiwan on a universal basis or not.
  • As pulmonary transplantation is another important factor potentially able to modify the natural history of the disease, data on patients diagnosed with IPF undergoing pulmonary transplantation during the study period should be considered in discussing the results.

Author Response

We would like to thank the editor and reviewers for their extensive review of our manuscript, as well as their important and helpful comments and suggestions. We provide our answers to certain comments below. The revised portions of the manuscript are highlighted in red color in our manuscript. We hope that the changes made will be considered satisfactory.

Reviewer 3 Report

This is a very interesting study where Yang and coauthors performed an epidemiologic analysis of idiopathic pulmonary fibrosis (IPF) in Taiwan covering the period of 2001-2011 years. The studies like this are extremely important for proper evaluation of morbidity, mortality, gender distribution, survival rate and risks of serious health/life threatening diseases like IPF.  

Major concerns.

  1. It would be beneficial for the study if authors would explain importance of using 2 definitions broad and narrow. Why the study of Ragu et all was used as an example for current study? What are advantages – disadvantages of usage of 2 definitions, what does it bring to the study? What approach bring more valuable data -broad or narrow – if authors can speculate/discuss it.
  2. Probably authors should mention that the severity of IPF was not considered in the current study, based on the limited availability in the database. Authors discuss this matter in the last sentence in the discussion section. However may be authors should consider include this important factor into the introduction section, because severity of IPF is an crucial factor for epidemiologic study, and since this factor couldn’t be considered from the beginning it need to be mentioned in the beginning of the manuscript.
  3. Since the study is operating with numerous numbers it would be very helpful for the readers, if authors could summarize all their final important numbers (prevalence, incidence, survival, age/gender etc into one table. And also, it would significantly make reading easier, if authors could include (in the same or in different table) comparison of their data with the data of epidemiology of IPF in Taiwan already published by other groups.
  4. It is strongly advisable for authors to update the supplemental tables a-c. As presented, the table have no titles, or explanation what data are shown in each column. The legend for these table also needs to be included into the supplemental material or methods section.

Minor concerns.

  1. The paper needs to be very thoroughly proof read for mistakes and missing spaces between the words, different font size (for example page 2, lines 63-80 of methods section.). All the above need to be unified though whole manuscript including the figure legends.
  2. The paper will significantly benefit from the extensive language correction.
  3. In the legend for figure 1 the definition for # symbol need to be specified.

Overall, this study is focused on important subject, and is valuable for the IPF research field especially for Taiwan and other Asian countries.

Author Response

(The authors gave the same response as above.)

Round 2

Reviewer 1 Report

I am satisfied with the revisions that have been made to the manuscript and recommend its publication.

Author Response

We would like to thank the reviewers for your review of our manuscript, as well as your important and helpful comments and suggestions. The revised portions of the manuscript are highlighted in red color in our manuscript. We hope that the changes made will be considered satisfactory.
